# ‘You’ve Just Got to Keep Pestering’: Barriers and Enablers of Attaining Continuity of Hepatitis C Care for People Transitioning Between Prison and Community Health Services in South-East Queensland, Australia

**DOI:** 10.3390/ijerph22020238

**Published:** 2025-02-07

**Authors:** Idin Panahi, Linda A. Selvey, Cheneal Puljević, Amanda Kvassay, Dorrit Grimstrup, Andrew Smirnov

**Affiliations:** 1School of Public Health, The University of Queensland, Herston, QLD 4006, Australiaa.smirnov@uq.edu.au (A.S.); 2Queensland Injectors Health Network, Bowen Hills, QLD 4006, Australia

**Keywords:** hepatitis C, prisons, health services for prisoners, qualitative research, social stigma

## Abstract

Highly effective direct-acting antiviral (DAA) therapies for hepatitis C (HCV) have been available in Australian prisons since 2016. To address treatment interruption following release from prisons, the Queensland Injector’s Health Network (QuIHN) launched a Prison Transition Service (PTS) in south-east Queensland, Australia. Presently, the factors associated with continuity of post-release HCV care are poorly understood. The objective of this qualitative study was to explore the barriers and facilitators to HCV treatment among people recently released from prisons among PTS clients and stakeholders. Qualitative interviews were conducted with 27 participants, namely, 13 clients and 14 stakeholders (health and community support workers) of the PTS. We conducted thematic analysis using the framework of person-, provider-, and system-level barriers and facilitators. Person-level barriers included competing priorities post-release, while facilitators included self-improvement after treatment completion, preventing transmission to family, and social support. Provider-level treatment barriers included enacted stigma, limited prison health service capacity, and post-release health system challenges. Systemic barriers included stigma relating to HCV, injecting drug use, incarceration, and limited availability of harm reduction services. Policy changes and investment are required to expand HCV treatment in south-east Queensland prisons to facilitate patient navigation into community care. In terms of reducing stigma among health staff and the general community towards people with HCV, a history of incarceration and/or who inject drugs is crucial for improving treatment rates. Strategies such as peer-led or nurse-practitioner-led models of care may help improve treatment completion. Continuity of HCV treatment post-release from prisons is essential for Australia to meet the WHO’s 2030 HCV elimination target.

## 1. Introduction

Hepatitis C virus (HCV) is primarily transmitted via blood-to-blood contact, and HCV infection can lead to cirrhosis and hepatocellular carcinoma [1]. The World Health Organization (WHO) developed a target for the elimination of HCV by 2030 [2], a goal adopted by many nations, including Australia [3]. The burden of HCV is disproportionately experienced by people who have experienced incarceration, largely due to the criminalisation of substance use [4,5]. In 2024 in Australia, there was an estimated HCV infection prevalence of 8% in national prison populations [6]. This makes prisons a vital setting and opportunity for initiating treatment and care for HCV [7]. Highly effective direct-acting antiviral (DAA) treatment, available within Australian correctional facilities since 2016 [5,8,9,10], has made HCV elimination in prison populations more viable due to shorter treatment duration and few side effects. However, substantial interruptions to the HCV treatment cascade occur in prisons, resulting in inconsistent screening and treatment [11].

Previous studies on DAA HCV treatment among incarcerated people have identified person-level, systemic, and social factors, which were considered barriers or facilitators to treatment [12,13]. The health care system can be considered an amalgamation of policy-, setting-, and provider-level interactions, and there is a further set of distinct systemic factors associated with prison health care [14]. This is particularly salient in Queensland, Australia, where prison health services are decentralised, and HCV treatment experiences may differ between prisons [15]. Understanding the influence of these factors on successful HCV treatment is necessary for developing appropriate and effective HCV treatment services and framing appropriate HCV prevention messages [16]. When a person is released from prison, they enter a new environment different from that experienced by people who have never been incarcerated, and vulnerability to drug-related harm increases, despite having greater access in community settings to harm reduction measures [17]. In the post-release space, people living with HCV are exposed to different interactions and experiences at the personal, provider, and system levels, which may confer new barriers which are currently poorly understood [18]. Developing policies and services which address aspects of the post-release environment may reduce this vulnerability.

Stigma and discrimination, operating at personal, systemic, and societal levels, have previously been identified as significant barriers to quality health care for people who inject drugs. Stigma, as described by Goffman (1963), is an “attribute that is deeply discrediting”, leading to a person being reduced “from a whole and usual person to a tainted, discounted one” [19]. Stigma has been further defined through its mechanisms, including enacted, anticipated, and internalised stigma [20,21]. The stigma experienced by people who inject drugs is further exacerbated by the experience of incarceration [22,23]. Although stigma has previously been identified as a barrier to treatment inside prisons, there is limited understanding of the specific effects of stigma on continuity of HCV care for people during transition between prisons and the community [24].

Presently, there is a gap in the research on the continuity of HCV treatment for people leaving prisons [25,26]. This qualitative study investigates the person-, provider-, and system-level barriers and facilitators to continuity of HCV care for people transitioning between prison and community settings in south-east Queensland, Australia, and it examines how processes and experiences within prison settings affect treatment continuity post-release. Our analytic approach is informed by the WHO Commission on Social Determinants of Health conceptual framework to define distinct categories of barriers and facilitators [13,27,28]. Within the framework, stigma is a product of power and social hierarchies, which, in turn, are shaped by systemic factors such as public policy, governance, and culture and societal values. System-level and person-level health determinants within the model are implicitly interlinked. Although there are studies on person- and system-level barriers to HCV care within prisons [13,25], this framework has not, to our knowledge, been previously applied to frame qualitative research related to experiences of HCV treatment during the transition from prison to the community.

## 2. Materials and Methods

### 2.1. Setting

This study was implemented in 2020 in conjunction with the Prison Transition Service (PTS) provided by the Queensland Injectors Health Network (QuIHN). The QuIHN is a community-led harm reduction and health service for people who use drugs, including case-managed HCV treatment [29]. The PTS aims to facilitate continuity of HCV treatment for people in transition into or out of men’s and women’s prison facilities in south-east Queensland. The PTS’s Prison Transition Worker communicates with clients and stakeholders before and after release, determining the client’s stage within the HCV treatment cascade and options for referral upon release.

### 2.2. Ethical Approval

Ethical approval was received from The University of Queensland’s Human Research Ethics Committee A (2019000780) and the West Moreton Hospital and Health Service Human Research Ethics Committee (HREC/2019/QWMS/54789).

### 2.3. Sample and Recruitment

Participants included clients and stakeholders of the QuIHN PTS. Stakeholders were health and community support workers identified by the Prison Transition Worker (PTW) as having worked with the service or having provided HCV treatment to people transitioning to or from prison facilities. A chain referral approach was used to purposively select different types of service providers, reflecting the range of professions and agencies involved in prison and transitional health services.

The PTW promoted the study to clients, facilitated organisation of interviews with interested clients, and provided a list of stakeholders interested in participating. Client interviews took place within three months of release from a prison facility, and where possible, interviews were scheduled adjacent to the client’s other appointments to reduce inconvenience. Interviews with participants were held privately at locations where clients of the PTS received follow-up HCV care. When client participants could only be interviewed via phone, the PTW provided the participant’s contact details, which were deleted following the interview or unsuccessful contact. Five client participants were contacted via phone. Only participants with at least a basic level of English comprehension were recruited, as assessed by the PTW. Prior to participating, written or verbal consent was obtained from clients and stakeholders.

### 2.4. Procedure

Semi-structured interviews were held with client and stakeholder participants. The interviewer began by administering informed consent and explaining the project in detail to potential participants. An interview guide was developed by members of the research team (Panahi, Selvey, Puljević, Smirnov), which was informed by a narrative literature review we completed on existing HCV treatment initiatives in prisons [13,17]. Open-ended questions explored clients’ experiences of HCV treatment in the transition period, including priorities after release, experiences of stigma, and aspects of care which they considered may have been obstructive or beneficial for completing treatment post-release. Stakeholder participants were asked to reflect on their observations of client experiences which fostered HCV treatment pathways post-release and experiences which inhibited continuity of care.

Clients were given the opportunity to not answer or leave the interviews at any point. They were advised that their participation or withdrawal would not affect their access to services. Client participants were also advised that they were allowed to have a friend, carer, or other trusted person stay with them during the interview for support. Interviews were voice-recorded with participant consent and were transcribed by an audio transcription service. All participating clients were provided with an AUD 50 supermarket voucher as a reciprocity payment. Transcripts were proofread for accuracy and de-identified by the primary author.

### 2.5. Analytic Method

Thematic analysis was used to identify themes in the data, using methods developed by Braun and Clarke [30]. As the authors had some preconceived understanding of the subject matter, a deductive approach was taken, and coding and development of themes was directed by pre-existing knowledge [30]. The first author began by carefully reading interview transcripts before coding the data in multiple-line chunks. The creation of similar codes was used to develop preliminary themes. The researcher iteratively analysed transcripts and preliminary codes to account for newly identified themes, and transcripts were inductively analysed at this stage. Double coding was carried out to provide more consistency in coding, and the final selection of themes was reviewed by the research team. Coding was performed using NVivo 12 Pro [31].

## 3. Results

A total of 13 adult clients of the PTS (*n* = 11 male, *n* = 2 female; C1–13) participated in interviews within three months of their release from prison. Four clients self-identified as Aboriginal. A total of 14 PTS stakeholders (*n* = 4 male, *n* = 10 female, S1–S14) were interviewed, with three stakeholders interviewed twice to explore the development of the PTS over the study period. Stakeholders included one general practitioner, two nurse practitioners, one clinical nurse consultant, three clinical nurses, four alcohol and other drug community health workers, and three community integration service workers.

The findings of this study are summarised in Figure 1, with the barriers and facilitators we identified grouped according to the personal, provider, and system levels. Corresponding elements of the WHO Social Determinants of Health Framework are also depicted. All the barriers and facilitators identified could relate to any stage of the HCV treatment cascade. Systemic-level factors are shown to influence both provider-level and personal-level factors, while provider-level factors influence barriers and facilitators at the personal-level. This aligns with the unidirectional aspects of the WHO Framework.

### 3.1. Personal Factors

#### 3.1.1. Competing Priorities After Release

Most stakeholders and clients recognised the typically busy and challenging circumstances faced following release from prison as a barrier to HCV treatment. As such, HCV treatment was often perceived by clients as a lower priority than other activities after release, such as finding stable accommodation, reconnecting with family, meeting probation and parole requirements, and finding employment:

C5 (Male):Ever since getting out, things have been hectic. Trying to get in… living in this society, when you’re… when everything… you’re in a routine society. Everything’s busy here. You’ve got kids going to school. You’ve got jobs to do. It’s hectic.

C2 (Male):I was worried about when I was getting out, I was worried about going to Centrelink, fixing up my shit, signing up my Job Network, finishing that shit. Go to probation and sign in…

#### 3.1.2. Stigma

Some stakeholders believed that clients’ internalised stigma deterred them from engaging in HCV treatment upon release and that their feelings of unworthiness or failure may make them reluctant to reinitiate treatment. Some clients noted experiences of enacted stigma related to their HCV status, injecting drug use, and incarceration, including disapproval and exclusion when back in the community. Other clients described that they avoided speaking about their infection with anyone due to anticipated stigma. Several clients and stakeholders explained how this stigma may lead to patients not seeking support for HCV treatment:

C3 (male):Yeah, I’ve been judged for going in jail before, yeah. For sure. [I’ve been judged in] all types of ways, all types of ways. Jobs, job interviews, and just going to interviews, and that. They just sit [you] down after and [they’re] ticking all the boxes, and bang, [they get to] the next three questions, you know? [Do] you have a criminal record, [have] you been in jail? And do you use drugs? And I say yes to all of them. And then they tell you you’re going to get a phone call that night, and that phone call never comes… (laughs) Never comes. Yeah.

C5 (male):Family doesn’t know. I haven’t told anybody. It’s just something that… Why worry them? Or why put the stigma on them? Because it is a contagious disease.

S3:I think that some people might have such a deep shame that can contribute to not actually accessing treatment and supports. You know, the common thing I hear from people is “I don’t want to waste taxpayers’ money, I don’t want to take that sort of cost off the government” … we have to really reiterate that they are a human being who deserves treatment just like anybody else.

C7 (female):… And [my ex-partner] just told me that his mum told him that I had the hepatitis C, that she wouldn’t let him use the shower at my place or eat at my place…

One client mentioned peer support as a facilitator to HCV treatment, as they described being more comfortable discussing HCV treatment with peers in prison than with health services, family, or friends due to shared experiences which helped them to not feel alone during treatment. Social support from friends and family members was also considered by clients to be a valuable facilitator for completing HCV treatment:

C5 (male):… it’s good to know I’m not alone. I wasn’t alone… It was good talking to them anyway, because it helped. It was just good…to talk to other people with the same issue. That you’re not alone. You’re not ostracised… I had a friend that went through treatment. The one that told me in jail to get my blood test and just be tested. I talk to him about it a little bit, because he went through the [same] program. And we still see each other out here. So he’s probably the only person, other than you guys, that are aware of it.

C6 (male):My dad and family have given me a lot of support… My parents know. But other than that, no one really…I don’t really explain [my HCV] to many other people except for family, except [some family members] that don’t [give me support] … because some of my family can be uptight.

#### 3.1.3. Self-Improvement and Family as Treatment Motives

Some clients stated that HCV treatment was an opportunity to improve their life circumstances after release, facilitating their treatment initiation and completion. Other clients’ concerns of transmitting HCV to family members motivated them to seek treatment during transition to the community. Of note is the fact that for one client, being re-treated for HCV alleviated her concern of being unable to see her children due to the possibility of dying from complications associated with cirrhosis, removing a major source of distress:

C7 (Female):Because, I guess [acquiring HCV] was the consequence of bad choices and I had the ability to now… To potentially turn that outcome into a better outcome for a bad choice. I had a second chance, I guess.

C2 (Male):[Treating HCV is important to me] ’cause I’ve got sisters at home. Little sisters, and my two little nieces …. And I cut myself shaving sometimes, and I don’t want to leave blood around and then they get Hep C. Know what I mean? That’s what I worry about. But I keep the blood away from them. Like, I tell them to just stay away from me…

C7 (Female):Oh, I know I don’t have [cirrhosis] now. But I worry about finding out I have cirrhosis of the liver or whatever, or not seeing my kids. So I guess one issue heightens the other issue. Because I don’t have my kids, so that heightens the thought of, “Oh my God, what if I get cirrhosis of the liver and die, and I haven’t seen my kids?” So to get treated again, it’s a real blessing in disguise for me. Because now I cannot think in that manner.

### 3.2. Provider-Level Factors

#### 3.2.1. Stigma

Stigma against people living with HCV was also observed as a provider-level barrier to seeking or completing HCV treatment. Stigmatising attitudes enacted towards clients by health or prison staff were described as occurring both in prison facilities and the general community. Stakeholders and clients described stigmatising attitudes of service providers towards people who had been imprisoned and had HCV or engaged in injecting drug use:

S2:[They will experience] stigma. Even non-prisoners. I hear very regularly how a GP has treated them. Once they’ve found out that they’ve got a positive result for [HCV], and being accused of being a drug user when they’ve never used.

C6 (male):To be honest, [health staff in prison facilities] don’t help you very much… They don’t really give you [any support]… I think that they would [treat me poorly], but more like I didn’t push myself to speak with them, in a way. I tried to keep myself segregated from them…

#### 3.2.2. Insufficient Information About HCV Treatment After Release

Some clients described how they were not appropriately counselled by prison health staff on HCV treatment before release, and while their pre-release consultations covered treatment processes, they provided little information about treatment after reinfection or treatment while continuing to inject drugs. This lack of information meant that these clients incorrectly assumed reinfection would lead to ineligibility for retreatment. Some clients stated they received no information on accessing testing and treatment post-release. This was a barrier to treatment in community and prison settings:

C4 (female):Well they could let, let girls know like where to go or where to start. ’Cause that’s part of the problem, like they tell us to go do all these things but we don’t know how… basically they said, “Oh, well you’re getting out soon so you can deal with it when you’re out”.

C7 (Female):I didn’t really think much about [reinfection]. [Failing treatment] was a bit disappointing at first, but… yeah. I didn’t really think much more of it. I didn’t know that I could get a second one until… I think it was [the prison nurse practitioner] called me up this time… I was just going to live with it… it is what it is.

#### 3.2.3. Nurse Practitioner S100 Prescribing

Drugs in Australia’s Highly Specialised Drug (s100) schedule were not able to be prescribed by nurse practitioners within prisons until April 2020. Due to this, staff were dependent on external or visiting medical officers for authorised prescriptions to be completed, which compromised the prison health unit’s ability to provide medication to patients in an efficient manner. The removal of this barrier has allowed nurse practitioner prescribing to be a critical facilitator in ensuring that patients are able to complete treatment in prison and after release:

S1:…It’s been, it’s probably one of the only real prohibitive factors in getting people into treatment sooner now, in the prisons… The main prohibitive factor is that s100 script. So, you know, [the nurse practitioner is] writing up all the scripts but [they] have to wait, usually one day a fortnight for the [senior medical officer] to then sign off on those and then send a pharmacy. So we’re losing at a minimum two weeks… [If nurse practitioners] had the right to prescribe DAAs in correctional facilities], [they] would able to script people daily like [they are able to] do in the community. … [they could] do all those scripts for those people that day, take them to the pharmacy the same day, if not the next day. So our pharmacy would be processing scripts for these people on a daily basis rather than fortnightly, or sometimes longer, basis.

S4:… [The West Moreton Nurse Practitioner] has worked up over 700–800 patients in the time that we’ve commenced that telementoring and he’s increased his knowledge from, you know, nothing to now being able to treat the cirrhotics and re-treatments, and really being quite experienced in Hep C… He just needs to be able to write a script.

S6:S100 prescribing it seems a bit silly. ’Cause, I mean, the thing is, for HIV and Hep B, I can see it, I can completely understand, they’re complicated diseases to treat with a whole raft of different medications …have complicated side effects. And they’re also chronic diseases that need chronic treatment, they’re not a short 3 month treatment that end up in a cure…It’s just ridiculous [that nurse practitioners can’t prescribe DAAs in prison], isn’t it?

#### 3.2.4. Correctional Officers

Several clients and stakeholders described interactions with correctional officers as a barrier to initiating HCV treatment. For example, clients needing appointments with prison health units for HCV testing, outside of routine testing upon entry, were required to request an appointment from a correctional officer. One stakeholder indicated that correctional officers may lack commitment to HCV treatment programs because of stigma towards people living in prison facilities and people who use drugs.

C2 (Male):It was frustrating me really bad. It was annoying me. I always have to get [the correctional officers] to call up medical and ask what’s going on. “Nah we’ll come and speak to you tonight”. They come back in a couple months.

S9:There are definitely people who don’t agree with [the HCV treatment program]. Corrections is an even harder sell. Trying to convince correctional officers that HCV treatment is a good thing… Correctional staff are probably not very tactful at [hiding] their disdain that… all this money is being spent on “these people”, but if the [people living in prison] hear that, I guess it impacts them. Whether it be shame, or that feeling of obstruction.

### 3.3. Systemic Factors

#### 3.3.1. Harm Reduction Strategies

Some clients discussed the lack of harm reduction strategies in prison settings, which presented as a notable barrier to HCV treatment seeking. Clients stated that people were less likely to engage in harm minimisation strategies in prison due to a lack of access to harm reduction services such as needle and syringe programs (NSPs) and opioid agonist treatment (OAT). The greater risk-taking behaviour in prisons (i.e., injecting drug use with shared needles) may reflect internalised stigma and low self-worth or pragmatic responses to aspects of custodial settings, including the prohibition of sterile injecting equipment. As re-exposure to HCV is likely within this context, knowledge of this likely re-exposure may lead some patients to consider treatment futile, even after release:

C3 (male):I don’t know, maybe, if I’m shooting up or something [post-release], and people that shoot up that don’t have Hep C, or maybe been in jail or something… they want to use your needle after you or something like that, well… you just decide to let [them know], as an honest person—“I’ve got Hep C”… then people look at you, “nah nah nah”. … but the people who have been in jail, I know, they quickly just take it out of your hands…

S4:Some of the prisoners don’t want to come for their final [post treatment blood test in the community] because they know that they’ve injected and probably re-infected themselves. … [we are trying to communicate that] we have no stigma or discrimination against this… we will treat you as many times as you need to be re-treated.

S1:…There’s a lot of women in there who are saying to me, “no, I don’t want to treat my Hepatitis C, what’s the point, I’m still using every day, I will not go on Hepatitis C treatment until you guys put me onto [opioid agonist treatment]”.

#### 3.3.2. Insufficient Capacity of the Prison and Community Health Services

Several clients and stakeholders mentioned how overcrowding and insufficient prison health care staff meant that many patients were unable to progress through the treatment cascade, thus presenting a substantial barrier to HCV treatment initiation and completion. Some stakeholders noted that the availability of community health professionals competent in treating HCV was inadequate for facilitating continuity of care:

C2 (Male):There’s not many nurses or doctors there…there’s like two nurses and two doctors there. And there’s like, the jail’s like all doubled on [over capacity].

S2:[HCV] was always treated in the specialty areas, so GPs have not been experienced to do it. We sent a survey out to, um, the GPs that we’ve been working with… We provided education. I’ve dropped education packs to all the GPs, and they’ve told us that they’re not confident to treat even with the education.

Many clients and stakeholders noted that the long delay between engagement with HCV services in prison and treatment initiation was a further barrier, likely due to the limited capacity of prison health services to meet demand. Other barriers to treatment completion included the time needed for diagnostic and prescribing processes and medication delivery before treatment initiation. Stakeholders indicated that long delays to treatment initiation contributed significantly to treatment disruption, and some clients were frustrated that they were unable to complete treatment while incarcerated as a result. This led to treatment failure, with patients needing to be retreated upon release:

C2 (Male):It was all fuckery, and they were fucking me around. Like, making me go [to the prison health service] after every four months or so. It was taking that long, like, to tell me, actually, if I got it or not. They kept stuffing me around. I didn’t know what’s going on [at any point]… I thought in my head “What’s going on?”…. Like, I got my Hep C test done, but I don’t even know—I don’t know if I’ve got it, or not. It’s been fucking, a couple months that I got tested—am I clean or not?

S1:… you’ve got a lag time between two to four weeks after that for medications to arrive at the jails to start treatment. So, hence why if someone’s going to be [released] in less than a month, and if I haven’t seen them yet, then it’s pretty much no chance of us giving them treatment…

C5 (Male):Yeah, I’d see the nurse practitioner twice, maybe, in prison. And after treatment started, I didn’t see him again. And I wasn’t going to talk to the prison staff about it, because it was pointless… They’d just say, “We’ll put your name down, and see someone”, and it never happens… You’ve got to push constantly. (Interviewer: What [do you think] is the reason why it wouldn’t happen?) C5 (Male): Whether they’re too busy. Whether they just don’t give a shit… It’s like that for everyone… You’ve just got to keep pestering and pestering. … It gets pathetic at times, the wait. Again, [there] could just be so many people.

#### 3.3.3. Responsibility for Navigating HCV Treatment After Release

Clients and stakeholders noted that the responsibility for ensuring treatment continuation, including health system navigation, rested primarily with clients, thus representing a significant barrier to treatment re-engagement post-release. This was due to uncertainty of who to contact to manage treatment continuation, in addition to previously noted post-release challenges:

C13 (Male):If you guys aren’t coming to help us, that whenever I’m released… if you guys don’t touch base with us after we get our lives back, we’ll just, we’ll just have [HCV] forever. Because there’s no way to get rid of it without the pills.

S4:… [Making appointments for HCV treatment is] generally not [the patient’s] priority once someone’s released from here. Making it so it’s up to them to phone us, you know, that’s a barrier in itself… It’s all up to them to push and make this happen.

S1:… Often, [people recently released from incarceration] don’t necessarily know who to go to get a script, or they still think they might need to go to a liver specialist when you can just get a script from a GP now.

#### 3.3.4. Unexpected Release or Transfer

Several stakeholders noted that patients’ unexpected release before or during treatment, whilst welcome, was a barrier to continuity of HCV care. Stakeholders noted that poor availability of post-release contact details led to missed opportunities for treatment:

S1:… Quite often we get people who were waiting for medications to be dispensed, and they’ve suddenly been released. Some court order parole or something and they’ve been released, so then the man’s medication will sometimes be sent to the prison and they’ve already gone…that’s a lost opportunity for that patient… [Some prison facilities] can be very, very low [in the proportion of patients completing treatment], because [the patients are] just constantly going to court, getting released, or getting transferred and you’re losing touch with them.

S4:We often don’t know when prisoners are getting out, so when we go and do their next appointment, that’s when we find out they’re getting out. We have tried to overcome that by talking to the prisoner about when they think they might be getting out, and do they have an address that [they] give us consent to finding them so that we can continue to get them engaged in care. But that doesn’t always work. Corrections [are] very difficult, they will not give me any contact details for patients once they’re released.… so I’m sort of at the mercy of the patient getting in contact with us and giving us updated contact details.

## 4. Discussion

This study addresses a gap in the knowledge of HCV treatment barriers and facilitators within the context of the transition from prisons to the general community. Our qualitative research identified person-, provider- and system-level barriers and facilitators to continuity of HCV care for people transitioning between prisons and the community in south-east Queensland, Australia. Stigma was commonly identified as a barrier to treatment at the person, provider, and system levels. Key barriers to continuity of care included delays to treatment associated with limited staff availability (which were frustrating for patients) and the complexity of post-release health system navigation. Some important facilitators were also identified; clients and stakeholders considered nurse-practitioner-led models of care as useful for facilitating HCV treatment in prison settings. Furthermore, several clients stated that preventing HCV transmission to family members was a source of motivation for completing treatment. Being free from hepatitis-related stigma by being cured was another client-reported motivation for completing treatment.

### 4.1. Person-Level Factors

We found that competing priorities after release from prisons was a key barrier to HCV treatment continuation and completion. Our findings reflect a broader evidence base showing that community settings for people recently released from prison are inherently challenging and difficult to navigate [17,18,32,33]. Both clients and stakeholders in this study reported that gaining access to housing and employment were higher priorities than HCV treatment for people after release. Similarly, the burden of engaging with parole or social security services was a barrier to treatment-seeking. Whilst these factors were presented as person-level barriers within our findings, they may be considered to result from the intersection of person- and system-level issues. These challenges are compounded by the fact that the responsibility for post-release service navigation is placed primarily with the patient [34]. These issues present opportunities for HCV prison transition programs that have a holistic approach and can provide support (e.g., active referrals) to address health system challenges for patients [33,35,36].

Both clients and stakeholders noted that social support was identified as a person-level facilitator to HCV treatment. Clients found discussion of HCV treatment easier with people with shared life experiences, such as peers in prison, compared to providers, highlighting the potential opportunity for peer support to increase rates of HCV screening and testing in prison and transitional settings. Strategies involving peer support workers have demonstrated effectiveness in improving HCV service engagement and treatment completion in vulnerable populations, including in prison settings [37,38,39]. For example, a study in Ireland found that employing incarcerated HCV peer educators to refer and accompany patients to the prison health service was feasible and effective for promoting engagement with HCV screening and treatment [38].

Some clients suggested that HCV treatment could help them become “a better person”, referring to improving their health while also becoming free of stigma, a perspective described in previous studies [13,23]. Clients in this study identified returning to friends and family as protective factors for completion of HCV treatment, but they also reported avoiding discussing HCV with friends or family due to stigma. Stigma experienced from family members may be challenging for people living with HCV, as families are an important source of love and support [40,41]. Without other support structures, stigmatisation from family may harm the patient’s fundamental human need to belong [40]. Reluctance to communicate with support networks due to stigma, and subsequent lack of support, may hinder engagement with HCV services and treatment completion and undermine wellbeing [35]. Promoting post-release community health strategies that include measures for reducing stigmatisation, including peer support roles and culturally appropriate service provision, may improve engagement with HCV support in these populations [33].

### 4.2. Provider-Level Factors

Clients and stakeholders reported that health staff and prison officers enacted stigma toward HCV patients and described ways in which this stigma created treatment barriers. These findings are consistent with evidence from community and prison settings indicating that stigma among people living with HCV is a significant barrier to treatment [24,42,43,44]. Stigma can become self-reinforcing, as stigma enacted on people can become internalised, with anticipation of subsequent stigmatising experiences leading patients to distrust or avoid interactions with health services [45,46]. Some client participants felt unworthy of treatment, and this feeling mirrored the disdain that some prison officers reportedly had for the patients of prison health services. Addressing this stigma and distrust of health services is likely to require initiatives at the provider and system levels due to the inherent nature of stigma [46]. Stigmatisation has been previously described as a “process linked to competition for power and the legitimisation of social hierarchy and inequality” [45]. There is a clear power imbalance between people who have been incarcerated and prison health staff and prison officers. This could be seen in the client participants’ responses indicating that prison officers’ role in organising appointments led to lengthy delays to treatment initiation.

Due to the changes to the Highly Specialised Drug s100 program in Australia allowing nurse practitioners to prescribe DAA medication in prisons, nurse-led models of HCV care are now more viable, with substantial improvements to efficiency and timeliness of treatment. Within this study, stakeholders noted that the removal of the s100 prescribing barrier would allow for patients to be given prescriptions for DAA medication daily, rather than having to wait for periods of up to several months. As release from prison was often noted to be unexpected, the ability to provide medication with shorter waiting periods may reduce attrition from the treatment cascade and may lead to higher rates of patients completing treatment with appropriate support from community services upon release. The removal of the need for referral to external or visiting specialists may lead to prison health units not being restricted by the availability of other health professionals or organisations, allowing for daily or weekly medication prescribing rather than on a fortnightly or monthly basis. Through the removal of these restrictions, it may be possible for nurse-led models of care to become substantially more efficient, as the requirement of visiting medical officers or the need for telehealth consultations with specialists may be significantly reduced, particularly in settings where assessments such as transient elastography or rapid HCV antibody or RNA testing can be carried out on-site. Previous studies have identified that nurse-led models of care are a feasible, effective, and efficient approach to providing HCV treatment through the supervision or support of a specialist using telemedicine [10,47]. An Australian study demonstrated that nurse-practitioner-led models in prisons can support a decentralised and patient-centred approach that increases HCV treatment rates with minimal need for specialist input [48]. As nurse-practitioners further reduce the need for specialist support, nurse-practitioner-led models of HCV care in prisons may be highly effective, and further research of such models is recommended.

### 4.3. System-Level Factors

Stigma was also observed to be a system-level barrier to life post-release, with one client stating that stigma was detrimental in his struggle to find employment. The stigma enacted on clients during interactions with a range of people in their social environment is shaped by broader societal influences [41,49]. People who have experienced incarceration and are living with HCV may experience intersectional stigma and discrimination associated with multiple circumstances: history of incarceration, history of injecting drug use, and stigma associated with bloodborne diseases [23,50,51]. In addition to these sources, Aboriginal and Torres Strait Islander peoples (First Nations Australians) also experience discrimination due to racism [41]. Our findings illustrate how these intersectional experiences of stigma reflect social hierarchies and processes of social exclusion, including denial of access to resources and acceptance within the community [19]. Community-based organisations and health services may be valuable in reducing barriers associated with intersectional stigma, and further research of post-release HCV treatment services is recommended [52].

One-third or more of our client sample were Aboriginal and Torres Strait Islander people, which reflects the hyperincarceration of these population groups [53] and health inequity with regard to levels of exposure to HCV [54]. Participants did not speak directly about racism; however, the intersectional experiences of stigma among Aboriginal and Torres Strait Islander peoples may have a “compounding” effect indistinguishable from experiences of racism [55]. The development of culturally safe and appropriate HCV treatment services, including peer-led services, is required to promote greater health equity for Aboriginal and Torres Strait Islander people living with HCV.

Stakeholders noted the absence of needle and syringe programs and lack of access to opioid dependence treatment within prison facilities. Further, both stakeholders and patients noted the perceived futility of getting treated due to the risk of reinfection in the absence of access to sterile injecting equipment [34,56,57]. Clients and stakeholders described cases where treatment was refused by patients in prison and post-release unless they were either no longer using substances or they were provided with opioid dependence treatment. Readily available access to opioid dependence treatment and the provision of needle and syringe programs in prisons would likely promote HCV treatment rates and equivalence and continuity of health care between prisons and general community settings [34,58,59,60,61]. Adequate prison health service capacity is required for patients to access HCV treatment. Clients and stakeholders identified insufficient staff availability and long wait times as barriers to patients initiating treatment, while the absence of services to help navigate treatment upon release compromised patients’ likelihood of treatment completion. Combined, these barriers impede the provision of comprehensive and holistic HCV treatment. Consequently, people who have experienced incarceration do not have equivalence of HCV care or health outcomes compared with people who have not been incarcerated [62]. Though access to opioid dependence treatment has been expanded into all Queensland prisons since 2022 [63], the absence of needle and syringe programs and limited access to health services is a violation of the United Nations’ Standard Minimum Rules for the Treatment of Prisoners, also known as the Nelson Mandela Rules [64].

#### 4.3.1. Limitations

This study is subject to limitations. This study took place in south-east Queensland Australia, and as such, it may have limited generalisability to other settings. As recruitment for this study was carried out by the PTS, client participants may have believed that negative responses regarding HCV treatment during transition would affect their access to other QuIHN services. This recruitment process may have also led to bias in selecting clients who were more open, engaged, or positive about the service. The researcher made efforts to reduce this potential response bias by reassuring participants of the confidentiality of their interview responses. The basic English language requirements for participant eligibility may have led to potential selection bias and excluded people from non-English-speaking backgrounds. We were only able to recruit participants already engaged with HCV treatment and clients of the QuIHN PTS, thus excluding people who were not linked to the service and limiting exploration of the experiences of individuals suffering from a lack of continuity of care [65]. Additionally, initial referrals to the PTS were for patients who had either completed HCV treatment, filled prescriptions for medication, or initiated treatment prior to release. Consequently, these findings do not reflect experiences of patients at the screening or assessment stage of the treatment cascade. The inclusion of only patients who were engaged with the PTS may have led to selection bias. The interview schedule did not include questions asking directly about experiences of racism. This may have limited the opportunity for exploration of these experiences. Furthermore, most study participants knew their release date in advance, allowing for efforts in linkage to care to be made before release, potentially leading to fewer treatment barriers. However, stakeholder participants were able to provide observations on the barriers experienced by people who were released with little notice, which was identified as a barrier to HCV care during transition.

#### 4.3.2. Implications

The findings of this study further highlight that people who have experienced incarceration are a priority population for HCV treatment and that the relative weakness of HCV treatment and prevention strategies in post-release settings may be contributing to poor continuity of care. Policy or health services which address this gap will be required to meet the WHO 2030 HCV elimination goals. Services which facilitate the transition of HCV care between prisons and the community may mitigate the effects of both prison and community setting barriers [33]. These services, alongside other community reintegration services, may also reduce HCV risk after release through linking people to harm reduction services. Ideally, HCV transition services should be developed in collaboration with the communities they aim to serve [66] and with a particular focus on lowering the minimum number of visits required to initiate treatment [67]. Providing opportunities to contribute to service development through co-design may be empowering for the patients who are accessing these services and provide a sense of ownership of health programs and opportunities to address stigma [25,37]. There have been some efforts made to use co-design approaches for HCV treatment in prison populations in Australia and the USA; however, these initiatives have been largely focused on education and advocacy [68,69]. Experiences from community settings suggest that the use of co-design and participatory processes can support the development of novel health services that dismantle existing provider–patient power imbalances [70]. Within prison and transitional settings, such initiatives could include the establishment of properly remunerated HCV peer support roles; client participants spoke about the tangible benefits of peer support for treatment completion. The development of co-designed services across prison systems may improve HCV treatment rates among people who experience incarceration.

Whilst targeted interventions from health services may alleviate some experiences of stigma during prison and community health system interactions, broader policy change is required to significantly reduce the effects of stigma as a structural determinant of health for people who have been imprisoned [50]. Policy changes to reduce stigma could include the decriminalisation of drug use, the provision of greater access to affordable housing for people who use drugs, and measures to reduce poverty [28].

## 5. Conclusions

This study examined the barriers and facilitators to HCV treatment following release from prison, as observed by people who were recently released from prison and relevant stakeholders. This study uniquely contributes to the literature on HCV treatment for people who have experienced incarceration, as the evidence related to post-release barriers and facilitators to HCV treatment continuity is limited. Stigma was noted as a key barrier across person, provider, and system levels. Additional person-level barriers included competing post-release priorities. At the provider level, inadequate counselling on treatment continuation post-release was considered a barrier. System-level barriers included insufficient or absent harm reduction services in prisons and the limited capacity of prison health services to meet the demand for HCV treatment.

Direct-acting antiviral therapies have made HCV elimination feasible, but increased rates of treatment and cure are needed for people living with HCV who have experienced incarceration. For HCV treatment and prevention strategies to succeed, treatment rates must increase significantly in this population. Treating HCV in prisons provides opportunities for engaging with populations who are difficult-to-reach or find treatment difficult-to-access. It is vital to ensure that release processes do not disrupt treatment. Resolving system-level issues may help address some of the social-level barriers such as stigma that can lead to treatment avoidance. Interventions such as peer-led approaches for HCV treatment may be helpful in promoting the continuity of HCV care, as peers were found to be valuable sources of information and support. The provision of harm reduction services in prisons and the streamlining of post-release health system navigation are examples of actions which are necessary for eliminating HCV in Australia and elsewhere.

## Figures and Tables

**Figure 1 ijerph-22-00238-f001:**
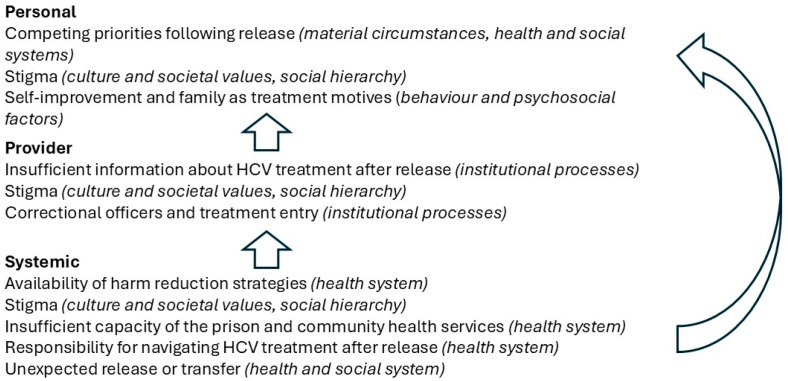
Identified barriers and facilitators Note: Text in italics shows the corresponding elements of the WHO Social Determinants of Health Framework. Barriers and facilitators may be applicable to any stage of the HCV treatment cascade.

## Data Availability

The data presented in this study are not available on request due to privacy and ethical restrictions.

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
