# Peer review of "‘You’ve Just Got to Keep Pestering’: Barriers and Enablers of Attaining Continuity of Hepatitis C Care for People Transitioning Between Prison and Community Health Services in South-East Queensland, Australia"

_ijerph, 2025, doi:10.3390/ijerph22020238_

Round 1

Reviewer 1 Report

Comments and Suggestions for Authors

Dear author, after reviewing the article entitled: You’ve just got to keep pestering’: Barriers and enablers of at- taining continuity of hepatitis C care for people transitioning between prison and community health services in South-East Queensland, Australia, I am sending you my suggestions and observations on the matter.

Regarding the structure of the article, the abstract is missing some elements such as: study objective, main result and main conclusion. The keywords must refer to desh or mesh terms related to the research topic. In the discussion, there is a repetition of the data described above in the results and although in some cases the results are compared, it is necessary to include the discussion of each of the aspects considered in the interviews such as personal factors, stigmas, provider level factors, etc. and to discuss its results with other similar studies. In the conclusions section, there is no answer to the objective about what are the barriers and facilities that the respondents described to continue with the treatment for hepatitis C.

That is all I can share with you.

Kind regards.

Author Response

Dear Reviewer 1,

Please see the attached updated manuscript and reviewer feedback table. Thank you for your comments and assistance in refining our work.

Kind regards,

Idin Panahi

Reviewer 2 Report

Comments and Suggestions for Authors

Abstract

1.       Need of the study needs to be strengthened = Include explanations to highlight the current gaps in HCV care during prison transitions in Queensland

2.       Mention the demographics of participants for better clarity.

3.       Incorporate specific examples of barriers and facilitators to make the findings more concrete.

Introduction

1.       The discussion on stigma is not presented well. All related aspects should be presents in a single paragraph for better comprehension.

2.       Once again, justify the need for the study by highlighting the lack of research on post-release HCV care transitions.

3.       Provide a brief explanation of how the WHO Commission framework informs this study’s approach.

Methods

1.       Any specific reason - why only clients with basic English proficiency were included? Scope for selection bias

2.       Provide more detail on the interview guide development, including specific questions or themes covered.

Results

1.       Highlight disparities or trends in the demographic data of clients and stakeholders.

2.       Include more diverse client quotes, especially from female or Aboriginal participants, to reflect varied experiences.

Discussion

1.       Discuss how overlapping stigmas (e.g., incarceration, drug use, and HCV) can be addressed holistically.

2.       Highlight the potential role of co-designing services with affected populations to improve outcomes.

References

1.       Add more recent references to strengthen the evidence base, particularly post-2020 studies on HCV care transitions.

Author Response

Dear Reviewer 2,

Please see the attached updated manuscript and reviewer feedback table. Thank you for your comments and assistance in refining our work.

Kind regards,

Idin Panahi

Reviewer 3 Report

Comments and Suggestions for Authors

The demographic characteristics of the participants in the study should be given in the text in a table. Was the sample selection made according to maximum diversity sampling?

Based on which criterion was the sample size decided?

In content analysis, data obtained through interviews, observations or documents should be analyzed in four stages.

(1) coding of data,

(2) finding codes, categories and themes,

(3) organizing codes, categories and themes,

(4) defining and interpreting the findings.

Validity addresses the accuracy of research results or the ability to solve the problem. In qualitative research, validity is the degree to which the researcher solves the problem he/she is addressing in the most impartial way possible.

The repeatability of research results is related to the concept of reliability. In other words, “could the same results be achieved if the study was conducted a second time?”

What was considered to ensure validity and reliability in the research? Should be written as a short title.

Author Response

Dear Reviewer 3,

Please see the attached updated manuscript and reviewer feedback table. Thank you for your comments and assistance in refining our work.

Kind regards,

Idin Panahi

Reviewer 4 Report

Comments and Suggestions for Authors

General Observations

  1. Structure Alignment
    • Some sections lack the granularity or emphasis necessary for the study objectives. For instance:
      • The results section does not sufficiently explore systemic barriers, which were highlighted in the introduction.
      • The discussion does not adequately propose actionable recommendations derived from the results.
  2. Missing or Weakly Addressed Aspects
    • The methods section does not clarify why only certain stakeholders were interviewed and whether this creates a sampling bias.
    • The limitations section does not fully explore the potential issues of generalizability beyond South-East Queensland or the inherent biases in participant selection.
    • Statistics are minimal, and qualitative rigor could be improved by using a triangulation framework to validate findings.
  3. Alternative Presentation Suggestions:
    • Results could be presented in a tabular format showing barriers and facilitators across person-, provider-, and system-levels. This would improve clarity and emphasize the connections between findings and study objectives.
    • A visual framework or flow diagram summarizing the barriers and facilitators along the hepatitis C cascade of care could enhance the discussion section.

Specific Feedback by Section

Abstract:

  • Suggestion: Include a brief mention of specific, actionable recommendations (e.g., scaling peer support programs, enhancing provider training) to make the abstract more impactful.

Introduction

  • Strengths: Clearly establishes the significance of the problem and highlights the importance of prison settings for HCV treatment.
  • Suggestions for Improvement
    • Add statistics on the prevalence of HCV in Australian prisons to ground the problem.
    • Clarify why South-East Queensland was selected as the study site and how findings could inform national or global strategies.
  • Provide specific regional data on HCV prevalence in prison populations to justify the study further.
    • Cite additional literature that discusses continuity of care frameworks in vulnerable populations.

Methods

  1. Ethical Considerations
    • Comment: While ethical approvals are mentioned, the manuscript does not address how researchers mitigated power imbalances between researchers and participants (especially clients who were recently released).
    • Suggestion: Add a description of measures taken to ensure participants felt comfortable sharing negative experiences about the program.
  2. Sample and Recruitment
    • Comment: The sampling criteria could result in biases as only those engaged with the program were recruited.
    • Suggestion: Include a justification for why non-engaged individuals were excluded and how this might limit the findings.
  3. Analytical Rigor
    • Comment: Thematic analysis is mentioned, but there is no detail on how inter-rater reliability was assessed.
    • Suggestion: Describe steps taken to ensure coding consistency across researchers, e.g., double-coding or peer review.

Results:

  1. Presentation of Results
    • Comment: Results are overly narrative and lack a structured summary.
    • Suggestion: Use a matrix or table to categorize findings under "person-level," "provider-level," and "system-level" barriers and facilitators.
  2. Statistics and Representation
    • Comment: The study does not quantify the frequency or distribution of themes (e.g., "X% of participants reported stigma as a barrier").
    • Suggestion: Use simple counts or percentages to add weight to qualitative findings.
  3. Specific Findings:
    • Comment: Structural barriers are discussed, but systemic enablers (e.g., supportive policies) are underexplored.
    • Suggestion: Identify and highlight any systemic strengths or best practices that could be scaled.

Discussion

  1. Alignment with Objectives
    • Comment: The discussion reiterates the results but does not fully explore their implications for policy or practice.
    • Suggestion: Explicitly link findings to the WHO 2030 HCV elimination goals and outline how these insights can inform policy changes.
  2. Actionable Recommendations
    • Comment: The recommendations are broad and lack specificity.
    • Suggestion: Propose clear, actionable steps for reducing stigma (e.g., mandatory training for prison staff) or improving transitions (e.g., funding peer navigators).

Conclusion

  • Comment: The conclusion reiterates points from the discussion without adding new insights.
  • Suggestion: Emphasize the unique contributions of this study (e.g., highlighting the role of peer support) and call for multi-sectoral collaboration.

Limitations

  1. Depth of Limitations
    • Comment: The manuscript briefly mentions limitations but does not critically examine their implications for the findings.
    • Suggestion: Discuss how the study's focus on engaged participants might have led to an overestimation of program success.

Author Response

Dear Reviewer 4,

Please see the attached updated manuscript and reviewer feedback table. Thank you for your comments and assistance in refining our work.

Kind regards,

Idin Panahi

Round 2

Reviewer 4 Report

Comments and Suggestions for Authors
  • Introduction & References

    • You have successfully integrated additional references post-2020 regarding HCV care transitions. Consider briefly highlighting any recent evaluations of nurse-practitioner-led models to reinforce the argument for s100 prescribing expansions.
  • Results

    • The results are well organized, and the figure has improved clarity. You may wish to note in the figure or caption that these barriers/facilitators apply across multiple stages of the HCV care cascade, for maximum transparency.
  • Discussion

    • The revised manuscript concisely addresses intersectional stigma, but readers might appreciate a short reference to any existing co-designed projects (if available) for continuity of HCV treatment in similar settings. This would lend even further support to your recommendation for peer-led or co-designed approaches.
    • Minor Edits
    •  
    • Throughout the text, watch for minor typographical or punctuation issues. Although these do not hinder comprehension, a final copy-edit could improve overall polish.
Comments on the Quality of English Language

The manuscript’s English style and clarity are now satisfactory, and minor editorial polishing can be done by the journal. There are no major grammatical errors that impede comprehension.

Author Response

Dear Reviewer,

Thank you for your feedback. Please see the attached document showing our responses to your comments.

Kind regards,

Dr Idin Panahi
